# Multi-Modulation of Doxorubicin Resistance in Breast Cancer Cells by Poly(l-histidine)-Based Multifunctional Micelles

**DOI:** 10.3390/pharmaceutics11080385

**Published:** 2019-08-02

**Authors:** Li Jia, Nan Jia, Yan Gao, Haiyang Hu, Xiuli Zhao, Dawei Chen, Mingxi Qiao

**Affiliations:** 1School of Pharmacy, Shenyang Pharmaceutical University, Wenhua Road 103, P.O. Box 42, Shenyang 110016, China; 2Department of Pharmacy, Heze Medical College, Heze 274000, China

**Keywords:** poly(l-histidine), pH sensitive micelle, MDR

## Abstract

Even though the reversal of multi-drug resistance (MDR) by numerous nanoparticles has been extensively studied, limited success has been achieved. To overcome this barrier, we report a rationally-designed pH-sensitive micelle, in which doxorubicin (Dox) and resveratrol (Res) were co-loaded. The micelle was based on methoxy *poly* (ethylene glycol)-poly(d,l-lactide)-poly(l-histidine) (mPEG-PLA-PHis), which integrated passive targeting, endo-lysosomal escape and pH-responsive payloads release. At a physiological pH of 7.4 (slightly alkali), Dox and Res were incorporated into the micelles core using the thin-film hydration method (pH-endoSM/Dox/Res). After cellular uptake, the micelles exhibited an enhanced dissociation in response to the acidic endosomes, triggering the release of Res and Dox. Furthermore, Res was observed to synergistically improve the cytotoxicity of Dox by down-regulating the P-glycoprotein (P-gp) expression, decreasing the membrane potential of the mitochondrial and ATP level, as well as inducing cell apoptosis mediated by mitochondria. The pH-endoSM/Dox/Res showed a prominent ability to decrease the IC_50_ of Dox by a factor of 17.38 in cell cytotoxicity against the MCF-7/ADR cell line. In vivo distribution demonstrated the excellent tumor-targeting ability of the pH-endoSM/Dox/Res. All results indicated that pH-endoSM/Dox/Res held great potential for the treatment of Dox-resistance breast cancer cells.

## 1. Introduction

Chemotherapy is still a treatment option for breast cancer, even though its efficacy has been seriously compromised due to the development of multidrug resistance (MDR), as well as the undesirable side effects of chemotherapeutic agents. Even though tremendous research effort has been made to overcome MDR, limited progress has been achieved in clinical studies [1,2]. One of the most challenging aspects in overcoming MDR is that multiple resistant mechanisms can be involved and probably even act jointly [3]. A number of resistant mechanisms have been revealed to contribute to the MDR, such as the ATP binding cassette (ABC) transporters [4], DNA methylation [5], glutathione S-transferase (GST) [6,7] and topoisomerase apoptosis [8]. Among them, membrane transporter-mediated MDR, (including P-glycoprotein(P-gp) [5,9], multidrug resistance-associated protein 1(MRP-1) [9] and breast cancer resistance protein (BCRP)) [10,11] is the most investigated mechanism due to the known clinical significance.

The elucidation of MDR mechanisms led to tremendous studies using various chemosensitizers (MDR modulators) co-administered with chemotherapeutic drugs by nanocarriers to combat MDR [12,13,14]. For example, various MDR modulators, such as Cremophor EL [15], Tween80 [16], TPGS [17] and Pluronic [18] have been used to combat MDR due to their inhibiting effect on the resistant mechanisms.

These MDR modulators have been elucidated to reverse MDR by sensitizing the resistant cancer cells to the cytotoxic drugs [19,20]. One of the most distinct advantages offered by the nanocarrier-based co-delivery systems is to enhance the selective accumulation in tumor and to synchronize the Pharmacokinetics profiles of different agents [21,22,23]. For example, a multifunctional micelle which possessed the targeting ability mediated by folate, effective endo-lysosomal escape and payload releasing modulation in acidic environments, was constructed to co-delivery Dox and Pluronic P85 to reverse the multidrug resistant [14]. Compared with normal micelles, the multifunctional micelles revealed enhanced cytotoxicity against MCF-7/ADR cells because of its capability in overcoming the biological barriers for intracellular payloads delivery [24,25].

However, most MDR reversal studies have been focusing on the reversal of the classical MDR mechanisms by using various MDR modulators [26]. This is probably one of the major reasons for the limited progress in combating MDR because of the failure of addressing the multifactorial nature of MDR. The MDR in the cancer cell was probably caused by more than one resistant mechanism. Therefore, future studies intended for MDR reversal are expecting to address more cellular MDR mechanisms by using more rationally-designed multifunctional nanocarriers to co-deliver MDR modulators and chemotherapeutics agents [27,28].

Resveratrol (Res), a natural phytoalexin obtained widely from the root extracts of *Polygonum cuspidatum* (buckwheat and knotweed of the Polygonaceae Family) and red grapes, has received significant attention in cancer chemotherapy [29,30,31]. Previous studies indicate that Res can overcome MDR by modulating apoptotic pathways, downregulating drug transporters and inhibiting cell survival signaling pathways, which make it a promising candidate as an MDR modulator [27,28,29,30].

In here, a multifunctional micelle was constructed by the copolymer of mPEG-PLA-PHis for the co-delivery of Dox and Res to exert a synergistic anti-tumor effect in MCF-7/ADR cancer (Scheme 1). The multifunctional micelles were designed to integrate passive targeting, endo-lysosomal escape and pH-triggered payloads release functionalities, in order to overcome the enhanced intracellular sequestration in endo-lysosomes, and to increase the intracellular concentration of payloads. Res was expected to combat multiple resistant mechanisms such as P-gp mediated efflux, over-expressed P-gp and the inhibition of cell apoptosis. The co-encapsulated Dox and Res micelles were prepared and characterized. The possible mechanisms underlying the superior efficiency of the micelles for MDR reversal was also investigated. Furthermore, the biodistribution of the micelles in vivo was evaluated using an orthotopic xenograft human breast cancer model.

## 2. Materials and Experimental Section

### 2.1. Materials

Doxorubicin (Dox) Hydrochloride was provided by the Zhejiang Hisun Pharmaceutical Co. Ltd. (Taizhou, China). Resveratrol (Res) was purchased from Guan Jie Biotech (Xian, China). Nα-CBZ-Nim-DNP-l-histidine was offered by GL Biochem, Ltd. (Shanghai, China). The mPEG2000, N,N′-Carbonyldiimidazole (CDI), Isopropylamine and verapamil (VRP) were obtained from Sigma-Aldrich (Shanghai, China). d,l-Lactide was purchased from Jinan Daigang Biomaterial Co Ltd. (Jinan, China). MTT, the ATP assay kit, the caspase activity test kit and the JC-1-mitochondrial membrane potential assay kit were provided by the Nanjing Jiancheng Bioengineering Institute (Nanjing, China). Lysotracker Green DND-26 and Hoechst 33258 were purchased from Dalian Meilun Biotechnology Co. Ltd. (Dalian, China). FITC-P-glycoprotein was purchased from BD Biosciences (Franklin Lakes, NJ, USA). The reagents used in cell culture were all offered by Solarbio Life Sciences (Beijing, China). Other reagents and chemicals, if not mentioned, were obtained from Concord Technology (Tianjin, China).

### 2.2. Tested Formulations, Cell Culture and Animals

Tested formulations: Dox solution; Res solution; Dox/Res: Dox and Res mixed solution (*w*/*w*: 1:1); pH-endoSM/Dox: endosomal pH sensitive micelles (methoxy *poly* (ethylene glycol)-poly(d,l-lactide)-*poly* (l-histidine) (mPEG-PLA-PHis)) loaded with Dox; pH-endoSM/Res: mPEG-PLA-PHis loaded with Res; pH-endoSM/Dox/Res: mPEG-PLA-PHis loaded with Dox and Res.

Cell culture: MCF-7 (human breast adenocarcinoma cell line) and MCF-7/ADR cell (Dox-resistant MCF-7), obtained from Nanjing KeyGen Biotech. Co., LTD. (Nanjing, China), were cultured at 37 °C in a humidified atmosphere of 5% CO_2_ after being cultured in Dulbecco’s Modified Eagle’s Medium (DMEM) and 1640 medium, respectively. Notably, to maintain the resistant property of the MCF-7/ADR cell line, 1 μg/mL Dox should be added into the medium.

Animals: Female BALB/c-nude mice (20 ± 2) g were obtained from the Department of Experimental Animals, Shenyang Pharmaceutical University (Shenyang, China). All animal experiments were carried out under the protocols approved by the ethics committee of Shenyang Pharmaceutical University (the project identification code: SYPU-IACUC-C2018-12-7-102, date of approval: 07/12/2018).

### 2.3. Synthesis and Characterizations of pH Sensitive Copolymers

The copolymer was synthesized according to the procedure described in our previous study [27]. The chemical structure was confirmed by ^1^H nuclear magnetic resonance spectra in CDCL_3_-d using Bruker DRX-600. The average molecular weights (Mw) and polydispersity index (PDI) of the copolymer was determined by the gel permeation chromatography (GPC) by Agilent 1200 series system.

### 2.4. Preparation and Incorporation Payloads into Micelles

The payloads were incorporated to the copolymer micelles by the thin-film hydration method [32]. First, Dox base was prepared by a dropwise addition of triethylamine (10 μL for per 10 mg Dox·HCl into the Dox·HCl aqueous solution (50 mg/mL) and extracted with CHCl_3_) [33]. Then 2 mg of Dox and Res (1/1, *w*/*w*) were added to 10 mg of mPEG-PLA-PHis in 10 mL of dichloromethane. The mixture was sonicated for 30 min. A thin film was obtained by rotary evaporation at 15 °С followed by hydrating with 10 mL of phosphate buffer saline (pH 7.4) (slightly alkali) for 20 min in order to obtain the payloads-incorporated micelles. The micellar solution was centrifugated at 9600× *g* for 10 min. The final micellar solution was collected by filtrating through a 0.45 μm membrane film.

Reference micelles were prepared by the same process, except that either 1 mg of Dox or 1 mg of Res was mixed with mPEG-PLA-PHis. The blank micelles were prepared with the same process, but this time eliminating the incorporation of payloads. DIR-incorporated micelles were obtained by mixing 0.1 mg of DIR with mPEG-PLA-PHis.

### 2.5. Characterization of Copolymer Micelles

The encapsulation efficiency of Dox and Res, respectively, were determined by an ultracentrifugation method [34]. The micelle solutions were placed into centrifuge tubes and centrifugated at the speed of 12,000 rpm for 10 min. Then, the supernatants were collected and passed through a 0.22 µm Millipore filter. The filtrate was transferred to a 10 mL volumetric flask and diluted with methanol to volume. The amount of free Dox and total Dox, respectively, were measured by a multifunctional microplate reader (Tecan, Austria) (Ex = 470 nm; Em = 585 nm), while the amount of Res was analyzed by high-performance liquid chromatography (HPLC) with a detection wavelength of 306 nm. The encapsulation efficiency (EE%) and drug loading content (DL%) of Dox and Res were calculated using the following equations
(1)$$\mathrm{EE}\%=\frac{m_{1}}{m_{2}}\times 100\%$$
(2)$$\mathrm{DL}\%=\frac{m_{1}}{m_{2}+m_{3}}\times 100\%$$
where, m_1_ is the weight of the encapsulated payload, m_2_ the weight of the feeding payload and m_3_ is the weight of copolymer.

Furthermore, the zeta-potential, particle size and its distribution of the pH-endoSM/Dox/Res micelle were characterized using dynamic light scattering (Malvern Zetasizer Nano ZS instrument, UK) subsequent to appropriate dilution. After the samples were prepared on the copper mesh by being negatively stained with 2% sodium phosphotungstate, the morphology feature of pH-endoSM/Dox/Res was captured by a transmission electron microscope (TEM, JEOL Ltd., Tokyo, Japan).

### 2.6. In Vitro Release of Dox and Res from Micelles

The dialysis method was used to investigate the Dox and Res release according to a previous report [35]. In brief, Dox, Res and pH-endoSM/Dox/Res were transferred into the dialysis bags (MWCO 3500 Da) and then immersed in 50 mL of phosphate-buffered saline (PBS) at different pH levels (0.01 M, pH 7.4, 6.5, 5.0), respectively. Then 0.5% (*w*/*v*) of Tween 80 was added in dissolution media to satisfy sink conditions. The system was kept at 37 °С under a shaking of 100 rpm. At each predetermined point, 2 mL of released medium was sampled and replaced with 2 mL of fresh medium. The amount of Dox and Res in the sample were assayed as described before.

### 2.7. In Vitro Cytotoxicity Study

MCF-7 and MCF-7/ADR cell lines (obtained from Nanjing KeyGen Biotech. Co., LTD.) were used to assess the cytotoxicity of various formulations in vitro [36]. MCF-7 (0.7 × 10^4^ per well) and MCF-7/ADR cell (1.4 × 10^4^ per well) were well seeded and incubated in 96-well plates with five replicates for one night. Then, Dox, Res, pH-endoSM/Dox and pH-endoSM/Dox/Res, respectively, were co-incubated with MCF-7 and MCF-7/ADR cells in 96-well plates at various concentrations for 48h. The original medium was replaced with the fresh medium (200 μL) containing 10% MTT solution and reacted with cells for another 4h. Finally, the medium was replaced with dimethyl sulfoxide (DMSO) (150 μL) to dissolve the formazan crystals formed in the viable cells. The untreated wells and untreated cells were used as blank and control, respectively. The optical density (OD) value at 570 nm was recorded by a microplate reader. The cell viability was calculated according to Equation (3). The resistance index (RI) of MCF-7/ADR cell was calculated according to Equation (4). The IC_50_ value of each sample was calculated by SPSS 17.0 (Chicago, IL, USA).
(3)$$\mathrm{Cell}\;\mathrm{viability}=\frac{{\mathrm{OD}}_{\mathrm{treat}}-{\mathrm{OD}}_{\mathrm{blank}}}{{\mathrm{OD}}_{\mathrm{control}}-{\mathrm{OD}}_{\mathrm{blank}}}\times 100\%$$
(4)$$\mathrm{RI}=\frac{{\mathrm{IC}}_{50}\left(\mathrm{MCF}7/\mathrm{ADR}\right)}{{\mathrm{IC}}_{50}\left(\mathrm{MCF}7\right)}$$

### 2.8. Intracellular Influx of Dox

To measure the intracellular influx of Dox, the flow cytometry analysis was performed. MCF-7/ADR cells (1 × 10^5^ per well) in logarithmic phase were seeded in 6-well plates with three replicates and incubated for 24 h. Then, PH-endoSM/Dox/Res (Dox concentration: 5.0 mg/mL) was added to the wells. The untreated wells were served as the blank and Dox and Dox/Res-treated wells were served as a control. After being incubated for 6 h, the cells were harvested, washed with PBS thrice and resuspended in 0.5 mL of PBS. The samples were detected by flow cytometry (BD Biosciences, San Jose, CA, USA) to quantify the internalized Dox.

### 2.9. Cellular Uptake and Intracellular Trafficking

Confocal laser scanning microscopy (CLSM) was used to investigate the cellular uptake and intracellular trafficking of Dox delivered by different formulations [37]. The Dox solution-treated group was used as a control. The MCF-7/ADR cells (2 × 10^4^ per well) were seeded on coverslips placed in a 6-well plate and cultured overnight. Dox solutions and pH-endoSM/Dox/Res (Dox concentration: 5.0 mg/mL) were added to the wells and incubated for 15 min, 30 min, 1 h, 2 h, 4 h and 6 h at 37 °С, respectively. After being washed with PBS thrice, the cells in the coverslips were treated with LysoTracker green DND for 30 min to visualize endo/lysosomes. Subsequently, the cells were washed by PBS to remove the excess stain, fixed with paraformaldehyde solution (4%, 0.01 M PBS), and stained with Hoechst 33258 for 10 min, followed by being captured using CLSM (Olympus FV1000-IX81, Japan).

### 2.10. P-gp Expression Determination

The P-glycoprotein (P-gp) expression of various formulations were detected by Flow cytometry [38]. The gene silencing efficiency was also detected by Flow cytometry. MCF-7/ADR cells (1 × 10^5^ per well) were seeded and cultured in 6-well plates for 24 h. Dox, Res, Dox/Res and pH-endoSM/Dox/Res were added and incubated for 6 h (Dox concentration: 5.0 mg/mL). Verapamil (calcium antagonists and multi-drug resistance (MDR) reversal agent) was used as a positive control. The cells were harvested, washed with PBS thrice and resuspended in 0.5 mL of PBS. Then the samples were detected after having been treated with 20 μL FITC-labeled P-gp monoclonal antibody at 4 °С for 30 min.

### 2.11. Mitochondrial Membrane Potential Detection

The fluorochrome JC-1(5,5′,6,6′-Tetrachloro-1,1′,3,3′-tetraethyl-imidacarbocyanine iodide) was used to detect the mitochondrial membrane potential l [39,40]. MCF-7/ADR cells (1 × 10^6^ per well) were planted in 6-well plates. After 24 h proliferation, Res, Dox, Dox/Res and pH-endoSM/Dox/Res were added to the cells for 2 h (Dox concentration: 5.0 mg/mL). The negative and positive control groups were the fresh culture medium and CCCP (Mitochondrial electron transfer chain inhibitor), respectively. After incubation for 6 h, JC-1 work solution (10 mg/mL) was added and incubated for another 20 min in dark at 37 °С. Then the cells were harvested and washed twice with JC-1 buffer solution. The fluorescence intensities of red (Ex = 488 nm, Em = 590 nm) and green (Ex = 488 nm, Em = 530 nm) were detected by a multifunctional microplate reader. The ratio of red to green was calculated and defined as the JC-1 fluorescence intensity ratio.

### 2.12. ATP Contents Assay

According to the ATP assay kit, the ATP content in the cells was determined by fluorescein-luciferase assay [18,34]. MCF-7/ADR cells (1 × 10^6^ per well) were planted and incubated in 6-well plates for 24 h. Then the cells were exposed to Dox, Res, Dox/Res and pH-endoSM/Dox/Res, and the culture solution was used as a negative control (Dox concentration: 5.0 mg/mL). After 2 h incubation, the cells were washed with pre-chilled PBS thrice and cell lysate was added. The supernatant was collected after centrifugation. Then, 50 μL of sample and 150 μL of ATP-monitoring working solutions were added and measured by a multifunctional microplate reader according to the operation manual of the ATP assay kit. The BCA kit was used to normalize the protein content and the ATP content in the samples was calculated by a pre-made standard curve.

### 2.13. Caspase Activity Assays

According to the caspase activity test kit instructions, MCF-7/ADR cells were planted and cultured in 6-well plates for 24 h, then co-incubated with Dox solution, Res solution, Dox/Res and pH-endoSM/Dox/Res for another 24 h. Blank medium was served as the control group. After being lysed and centrifuged at 9600× *g* for 1 min at 4 °С, the supernatants were collected and interacted with the peptide substrates of caspase 3, caspase 8 and caspase 9, respectively. The activities of caspase were calculated based on the absorbance values at 405 nm of reactants detected by a microplate reader.

### 2.14. Biodistribution of pHendo-Sensitive Micelles In Vivo

To investigate the biodistribution of pH-endo-sensitive micelles, the MCF-7/ADR cells were injected subcutaneously into the right axillary fossa of female BALB/c nude mice ((1 × 10^6^ per 200 μL PBS). When the tumor size reached 150–200 mm^3^, DIR-loaded micelles (0.1 mg/mL) were injected though the tail vein of the MCF-7/ADR xenograft nude mice. The Kodak In Vivo Imaging System FX PRO was used to image tumor-bearing nude mice with the wavelength at Ex/Em 720/790 nm after anesthesia. The time-dependent fluorescence distribution was detected at 0.5, 1, 2, 4, 8, 12, 24, 36, 48 h after injection. After 48 h, BALB/c nude mice were sacrificed. The major organs and tumors were collected under anesthesia for ex vivo fluorescence imaging.

To further verify the tumor targeting ability of pH-endoSM/Dox/Res in vivo, the tumor-bearing mice were injected with Dox solution, pH-endoSM/Dox and pH-endoSM/Dox/Res, respectively (Dox: 5.0 mg/kg). The mice were sacrificed after 48 h, and the Dox fluorescence signals of the main organs and tumor tissues were detected (Ex: 497 nm; Em: 588 nm).

### 2.15. In Vivo Antitumor Efficacy and Safety Evaluation

The MCF-7/ADR tumor-bearing mice model was established as described above. The mice were randomly divided into four groups (five mice per group) when the tumor grew to about 100 mm^3^. The four groups were intravenously injected with saline, Dox solution, pH-endoSM/Dox and pH-endoSM/Dox/Res, respectively (Dox: 5.0 mg/kg). After 48 h, these mice were sacrificed, and then tumor tissues and the main organs were harvested and weighted. The tumor inhibition rates (TIR%) were calculated by the following formula. Finally, the major organs and tumor tissue were fixed with 4% paraformaldehyde for hematoxylin and eosin (H&E) staining according the manufacturer information. Tissue sections were observed and imaged using an inverted microscope.
(5)$$\mathrm{TIR}\%=\frac{Wc-Wt}{\mathrm{Wc}}\times 100\%$$
where *Wc* and *Wt* represent the tumor weight of the control group and the treated groups, respectively.

### 2.16. Statistical Analysis

Quantitative results were all expressed as mean ± standard deviation. Statistical analysis was determined by one-way analysis of variance (ANOVA) and Student’s *t*-test among each group (SPSS 17.0 software, SPSS Inc.). A *p* value smaller than 0.05 was considered statistically significant.

## 3. Results and Discussion

### 3.1. Characterizations of mPEG-PLA-PHis by Proton Nuclear Magnetic Resonance (^1^H-NMR) and GPC

The nuclear magnetic resonance spectroscopy (NMR) spectrum of copolymer was presented in Figure 1A with CDCl_3_ as the solvent. The peaks at δe are at 5.24 ppm, δd at 3.67 ppm, δc 3.47 and δb being 1.83 ppm, were attributed to mPEG-PLA (−COCH(CH_3_) O−, −OCH_2_CH_2_O−, −OCH_3_ and −COCH(CH_3_) O−, respectively). The intensity ratio of PLA at δe = 5.24 ppm (−COCH(CH_3_) and mPEG at δc 3.47 ppm (−OCH_3_) represented the degree of PLA block [41]. Two typical chemical shifts appeared at δa 1.55 ppm (−C(CH_3_)_2_−) and δg 4.35 ppm (−CH−NH−) and indicated the successful connection of PHis block to the PLA block. Based on the intensity ratio of δa 1.55 ppm to δg 4.35 ppm, the polymerization degree of poly(L-histidine) was determined to be 6. As shown in Figure 1B, gel permeation chromatography (GPC) also demonstrated the successful synthesis of the copolymer with Mw of 5511 and a low PDI.

### 3.2. Characterization of the Micelles

The characterization results of pH-endoSM/Dox/Res are shown in Table 1. The EE% values of Dox and Res in the mPEG-PLA-PHis micelles were above 80% respectively, indicating a good payload encapsulation. The dynamic light scattering test of Dox/Res-loaded micelles showed average an particle size of 75 nm with unimodal distribution (Figure 2A). The Dox/Res-loaded micelles were spherical particles with an average size of around 50 nm, as demonstrated by transmission electron microscope (TEM) images (Figure 2B).

The release of Dox and Res from the micelles were investigated at different pH conditions simulating the biological pH (7.4) and acidic tumor pH (6.5, 5.5). Prior to conducting the release tests, the Dox and Res releases from the stock solution were tested. Non-encapsulated drugs were completely released within 4 h, indicating the free diffusion of drug molecules across the dialysis membrane independent of pH. As shown in Figure 2C, pH-endoSM/Dox/Res micelles showed burst release (~30%) in 4h, followed by a typically sustained release at pH 7.4 (slightly alkali).

The cumulative payload release only reached around 40% after 24 h, indicating the good stability and low payload leakage of the micelles. However, as the pH declined from 7.4 to 5.5 (from alkali to acidic), the cumulative payload released remarkedly increased to ~70%, as compared to that at pH 7.4 at 24 h. The pH-accelerated payload release was caused by the dissociation of the imidazole group of PHis blocks, which induced the disorder of micellar structure [35]. The dissociation extent of PHis blocks was closely related to the environment pH. As the pH decreased, more imidazole groups would become dissociated, resulting in a higher extent of dissociation and more severe micellar structure disorder [41]. Therefore, the micelles showed a much faster payload release at pH 5.0 than that at pH 6.5. The similar pH-triggered release characteristics of Dox and Res indicate the potential of pH-endoSM/Dox/Res as a carrier for the co-delivery of agents with different physiochemical properties and the manipulation of Pharmacokinetics/Pharmacodynamics (PK/PD) properties for achieving the synergistic effect.

### 3.3. Enhanced Cytotoxicity Against MCF-7/ADR

In advance of the cytotoxicity test, the resistance characteristic of the cells was first investigated. According to the IC_50_ values (inhibitory concentration to reduce cell survival to 50%), the resistance index (RI) of the MCF-7/ADR cells was calculated to be 62.67, demonstrating that the cells possessed strong resistance to Dox.

The cytotoxicity of different formulations against MCF-7/ADR cells was determined by the MTT method. As shown in Figure 3A,B, as the concentration of Res and Dox increased, the cell viability of the MCF-7/ADR cell were both sharply decreased, indicating that the cytotoxicity of Res and Dox presented dose-dependent property. The concentration of Res was set to 10 µg/mL for the following studies due to the moderate cytotoxicity at this concentration. The antitumor effect of Dox against resistant cancer cells could be further reinforced by the presence of Res (Figure 3C). The results showed that the Dox/Res at the ratio of 1:1 (*w*:*w*) exhibited the best synergistic effect. Therefore, this Dox/Res ratio was chosen for the following studies.

The blank micelles showed good biocompatibility as characterized by more than 90% cell viability (data not shown). The cell cytotoxicity and IC_50_ values of the pH-endoSM/Dox and pH-endoSM/Dox /Res micelles against MCF-7/ADR cells were shown in Figure 3C and Table 2. The pH-endoSM/Dox/Res showed much lower IC_50_ value and higher RF compared to Dox solution and pH-endoSM/Dox (*p* < 0.05), demonstrating that the cytotoxicity against the resistant cells could be remarkably enhanced by co-delivery of Res [31].

### 3.4. Mechanisms for Reverting MDR

#### 3.4.1. Cellular Uptake and Intracellular Dox Accumulation

Flow cytometry and CLSM were first used to examine the intracellular Dox accumulation. After co-incubation with different formulations, the fluorescence signals of Dox inside the MCF-7/ADR cells were recorded (Figure 4A). The moderate MDR reversal observed with pH-endoSM/Dox was attributed to the endocytosis pathway of the micelles which partially bypass the efflux effect caused by a series of efflux pumps such as P-glycoprotein (P-gp) and the pH-triggered payload release, leading to the increased cumulation of Dox and Res in the cells. Furthermore, co-delivered Res markedly increased the intracellular influx of Dox by 5.25-fold, indicating that Res was capable of inhibiting the efflux of Dox. Therefore, these mechanisms endowed the micelle with a higher intracellular Dox accumulation than the Dox solution [42].

Besides, CLSM was also used to investigate the internalization and subcellular distribution of Dox. For the sake of observation, Hoechst 33258 (nucleus selective dye, blue) and LysoTracker DND-26 (endo-lysosome selective dye, green) were utilized to stain the nucleus and endosome, respectively. After cellular incubation with pH-endoSM/Dox/Res and Dox solution for different time intervals, the fluorescence signals were captured and presented as microphotographs in Figure 4B. The pH-endoSM/Dox/Res group showed weak yellow fluorescence (overlap of Dox with LysoTracker) after 30 min of incubation, indicating that Dox did accumulate in endosomes after endocytosis. The gradually increased red fluorescence of Dox and yellow fluorescence in merged images with incubation suggested the gradual accumulation of micelles in the endosomes. As the incubation time reached 6 h, the pH-endoSM/Dox/Res-treated group showed the overlapped Dox signal with the Hoechst 33258 signal, which revealed an apparent purple fluorescence, indicating that Dox escaped from endosomes and distributed into the nuclei. The higher intracellular Dox induced by the mPEG-PLA-PHis micelle could be attributed to the endocytosis pathway of the micelles, an acidic pH-triggered payload release, as well as the co-delivery of Res that showed the MDR reversal effect [39].

#### 3.4.2. Effect on P-gp Expression

To further assess the cellular mechanisms for reverting the MDR, P-gp expression inside the MCF-7/ADR cells was investigated by flow cytometry with Ver (a competitive P-gp inhibitor) as a positive control. As seen in Figure 5, Res, Dox/Res and the pH-endoSM/Dox/Res micelles group showed significantly lower P-gp expression compared with our control group (*p* < 0.05), while this difference was not significant in the Dox group (*p* > 0.05), demonstrating that Res was capable of down regulating the P-gp expression for MDR reversal.

#### 3.4.3. Effect on Energy Metabolism Mediated by Mitochondria

P-glycoprotein (P-gp) pumps cytotoxic drugs out of cells with the help of ATP, leading to the reduced intracellular drug accumulation and cells resistance to the drugs [43]. The energy of P-gp comes from ATP that is produced by mitochondrial oxidative phosphorylation [44]. Therefore, the effect of the Dox/Res incorporated micelles on cellular energy metabolism in resistant cancer cells was investigated.

A JC-1 fluorescent probe was utilized to detect the mitochondrial membrane potential. The existing form of JC-1 varies with the level of mitochondrial membrane potential and presents different fluorescence which helps to visualize the membrane potential. At normal conditions, the potential of the mitochondrial membrane is relatively high and the JC-1 probe presents red fluorescence. When the potential decreases, it presents green fluorescence. The red/green fluorescence intensity ratio decrease indicates the mitochondrial depolarization (non-functional mitochondria).

As shown in Figure 6A, all formulation groups containing Res showed a significant difference (*p* < 0.05) in decreasing the mitochondrial membrane potential compared to the control group, indicating that Res could disturb energy production by mitochondrial.

To inspect this effect, the ATP level inside the MCF-7/ADR cells was further detected. It could be seen in Figure 6B, the pH-endoSM/Dox/Res group showed the strongest inhibition on the ATP level inside the cells (*p* < 0.05). The decreased mitochondrial membrane potential and ATP level in the cell caused by Res could inhibit the drug efflux mediated by P-gp, resulting in the enhanced intracellular accumulation of Dox.

#### 3.4.4. Effect on Cell Apoptosis

To investigate cell apoptosis initiated by the Dox/Res micelles, the activities of caspase in MCF-7/ADR cells were tested [39]. As illustrated in Figure 7, considerable enhancements of caspase 3 and caspase 9 activities could be found with Res, Dox/Res and pH-endoSM/Dox/Res groups, as compared to those in the free Dox and control groups, while no obvious change of caspase 8 activity was found in all groups. The results suggested that cell apoptosis induced by pH-endoSM/Dox/Res micelles was attributed to the Res-induced mitochondria-dependent signaling pathways. The disruption of mitochondrial membrane potential caused the activation of caspase protein [45]. Caspase 9 could be recruited and activated, resulting in the formation apoptosome complex, which also activated caspase 3, leading to the following cell death [46].

All of these results suggest that the micelle reversed the MDR by multiple mechanisms including a co-delivery of Dox and Res, an endo-lysosomal escape mediated by the PHis block, P-gp expression down-regulation and mitochondrial-dependent apoptosis, which produced the synergistic effect for MDR reversal.

### 3.5. In Vivo Biodistribution Studies

The in vivo biodistribution of pH-endo-sensitive micelles was tested by an optical image system. The micelles loaded at 0.1 wt % DIR were intravenously injected into tumor-bearing mice. As shown in Figure 8A, the fluorescence intensity of the mPEG-PLA-PHis/DIR micelles was gradually increased in the tumor region after administration. At 0.5 h post injection, most of DIR/mPEG-PLA-PHis micelles were accumulated in the liver. As time progressed, higher fluorescence could be detected in tumor than other tissues at 12 h post injection. The strong fluorescence could last for 48 h in the tumor site. The results indicated that the micelles had a long blood circulation property, facilitating the enhanced permeability and retention (EPR)-mediated tumor targeting. After capturing the last real-time image photo, the tumor tissue and major organs were excised for capturing the ex vivo images (Figure 8B). The strong fluorescence of DIR was observed in the excised tumors, confirming the targeting payload delivery ability of the mPEG-PLA-PHis micelles.

The biodistribution of Dox delivered by pH-endoSM/Dox/Res was further investigated with Dox solution and pH-endoSM/Dox as control. The Dox fluorescence signals of the main organs and tumor tissues were shown in Figure 8C. After 48 h administration, The pH-endoSM/Dox/Res exhibited more Dox accumulation in the tumor than the Dox solution and pH-endoSM/Dox. It is consistent with in vitro results that co-delivered Res markedly increased the intracellular accumulation of Dox. It is worth noting that two micelles exhibited lower Dox accumulation in heart than the Dox solution. Dox is known for its serious cardiotoxicity, which limits its clinical application [47].

The copolymer based on micelles were capable of effectively reducing the distribution of DOX in the heart than the DOX solution, offering an alternative to reduce the cardiac toxicity of Dox [48,49].

### 3.6. In Vivo Antitumor Efficacy and Safety Evaluation

In vivo antitumor efficacy was investigated at 48 h after intravenously administered different formulations. As expected, Dox solution showed little anticancer effect (Figure 9A), which could be attributed to the nonspecific distribution of DOX as well as the resistance of the cancer cells to DOX. pH-endoSM/Dox/Res displayed a much stronger tumor inhibition rate than pH-endoSM/Dox, presenting much lower tumor weight (Figure 9A,B). This was in accordance with the in vitro cellular evaluation, which could be attributed to the co-delivery Dox and Res by pH-endoSM for reverting MDR via multiple mechanisms such as the efficient delivery of payloads, down-regulating P-gp expression and inducing cell apoptosis. The images of H&E staining on tumor tissues (Figure 9C) exhibited that the pH-endoSM/Dox/Res treated group produced much larger areas of necrosed cells than other formulation-treated groups.

In addition, the safety evaluation of different formulations was further investigated by H&E staining analysis. The Dox solution group exhibited severe cardiotoxicity, hepatotoxicity and nephrotoxicity, as characterized by the significant hydropic degeneration, vacuolar degeneration and spotty necrosis. In contrast, the pH-endoSM/Dox and pH-endoSM/Dox/Res group exhibited much less pathological change in major organs, indicating the excellent safety.

## 4. Conclusions

In summary, a multifunctional micelle based on the pH sensitive copolymer of mPEG-PLA-PHis was constructed co-deliver Dox and Res for multi-modulation of doxorubicin resistance tumor cells. The micelles demonstrated excellent encapsulation of Dox and Res as well as the relatively low leakage of payload at the biological condition. The pH-endoSM/Dox/Res micelles that were showed triggered the payloads release because of the protonation of PHis blocks in the micelles under acidic pH. The co-delivery of Res and Dox with the micelles exhibited the significant synergistic effect and enhanced cytotoxicity against MCF-7/ADR cells. The cellular distribution of Dox indicated that the Phis-based copolymer facilitated the effective endo-lysosomal escape of Dox and the translocation to the nucleus. The pH-endoSM/Dox/Res micelles were found to reverse MDR in a multiple way, including enhancing the intracellular accumulation of Dox by copolymer-facilitated endosomal escape, and down-regulating the expression of P-gp, reducing the membrane potential of the mitochondrial and declining ATP level inside cells, and triggering mitochondria-dependent cell apoptosis. The pH-endo-sensitive micelles exhibited more Dox accumulation in the tumor and less Dox distribution in the major organ, helping to enhance anti-tumor efficacy and reduce the Dox-related toxicity. The pH-endoSM/Dox/Res micelles have been demonstrated as a potential delivery system to reverse the MDR.

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
