# Peer review of "Multi-Modulation of Doxorubicin Resistance in Breast Cancer Cells by Poly(l-histidine)-Based Multifunctional Micelles"

_pharmaceutics, 2019, doi:10.3390/pharmaceutics11080385_

Round 1
Reviewer 1 Report
The manuscript is well written and should be accepted in the present form after moderate language editing. The authors should take help from a professional English editing service.
Reviewer 2 Report
The manuscript entitled "Multi-modulation of doxorubicin resistance in breast cancer cells by poly(L-histidine) based multifunctional micelles" describes a very important and current topic. Cancer respresent one of the leading causes of death in developed countries, and MDR is responsible for increased mortality and decreased quality of life for the patients. However, I have serious concerns regarding the current status of this manuscript.
Generally, the setup/organization of the paper is good, however, there are serious flaws in English Grammar and Scientific Vocabulary.
A. Introduction:
Line 32: mainstream method= unscientific term
Line 33: avalilability = bioavailability?
Line 36-38: there are a lot more resistance mechanisms, this is just a VERY select group.
Line 39: MRP1, BCRP etc... should have been mentioned too.
Line 50-77: the choice of model systems, verapamil, resveratrol, co-polimer etc. is TOTALLY artbitrary, and the choice is not explained whatsoever in the introduction section (or just very briefly).
I feel that there are many more references that could have been included.
B. Materials and methods: this section is plagued with minor inconsistencies. There are practically no references in this section? are all of these methods brand new?
In addition, most of the methods are not explained in a way, that would allow others to reproduce them-this is the basic tennent of scientific publication!. Are they credible at all? (especially 2.8.-2.14.)
Controls are missing in most of the experiments. I feel that this negates all siginificance of the results, as there is no vantage point/comparison for the reader.
There are a lot of unnecessary abbreviations in the paper, most in the MM section, which makes reading extremely difficult.
2.1.-2.2. The same issue, the choice of model systems is not explained.
2.3.-2.6. is fine
2.7. MTT is a redox-active compound. Did the authors check beforehand, whether the micelles/DOX/Res react/induce colour change in MTT in these concentrations, without the cells?
Positive/negative controls? You have nothing to compare to...
2.15. Software?
Results
The majority of results are adequately presented and described, however, the reviewer has a hard time believing the validity of these results, due to flaws in experiental desing.
Conclusions
There is no discussion of the results in light of results available in the literature, which is very troubling. In addition, the conclusion section is also quite short for a paper using 10-12 different methods.
Reviewer 3 Report
The manuscript "Multi-modulation of doxorubicin resistance in breast cancer cells by poly(L-histidine) based multifunctional micelles" by Li Jia is an intersting article. The topic of the study is important and have a potential to contribute to the field of the clinical therapy. The article is potentially valuable to public health experts and physicians.
I believe that editing and rewriting same part of the text is necessary to make the manuscript easily readable, but this effort would be well worth it as the authors have included a great deal of information.
However, I have following comments that should be addressed:
- I recommend that a native speaker of English review the manuscript to improve word choice, sentence structure, and grammar.
- the mathematical process should be reexplain is to technical written, need to be write simple and clear.
-The paragraph between row 285-297 can be reexplain.
- The conclusion part need more explanation and clear points.
- Please recheck the References order
Author Response
1. I recommend that a native speaker of English review the manuscript to improve word choice, sentence structure, and grammar.
The authors’ answer: Suggestion has been taken. Our manuscript has been reviewed by a native English speaker to improve readability. These changes have been highlighted in the separately submitted revised manuscript.
2. the mathematical process should be reexplain is to technical written, need to be write simple and clear
The authors’ answer: As suggested by the reviewer, we have carefully revised and reexplain the mathematical process in the manuscript to make it simpler, which was presented in the revised manuscript (line147 – line148.)
3. The paragraph between row 285-297 can be reexplain.
The authors’ answer: Suggestion has been taken. The paragraph between row 285-297 have been revised carefully (line 305 – line317).
4. The conclusion part need more explanation and clear points.
The authors’ answer: As suggested, the conclusion part of the manuscript has been carefully revised. (line 446- line 461).
5. Please recheck the References order.
The authors’ answer: Thanks to the careful check and warm reminder. We had examined the References order in the manuscript thoroughly, and the corresponding corrections have been made in the revised manuscript.
Reviewer 4 Report
In the present study Li Jia et al. have designed a pH sensitive micellar system with doxorubicin (Dox) and resveratrol (Res) evaluating in vitro the uptake and the release of Res and Dox in response to the pH changing. Furthermore, stronger cytotoxicity of the system compared to Dox alone was demonstrated against MCF-7/ADR cell line. Mechanistically, this effect is mediated by down regulating the P-gp expression, decreasing membrane potential of mitochondrial and ATP level, and finally cell apoptosis mediated by mitochondria. On the other hand, in vivo distribution demonstrated excellent tumor-targeting ability of the system. The authors conclude that these effects can improved the efficiency of Dox treatment and resistance against breast cancer cells.
Overall, the paper is well written, the experimental design is simple but very clear and the results are concordant with the conclusions. Furthermore, the system is potential interesting for future implication in the treatment for cancer but an important limitation of Dox treatment appear to be not evaluated. In fact, there are some major and minor points to be resolved before considering it suitable for publication.
Major Points:
1. The most important complication in the use of Doxorubicin is the toxicity induced in non-cancerous cells. In fact, the authors show in vivo the bioavailability of the micelles predominantly only in the tumor and not in the other organs. In any case, the authors must show in vitro experiments, of tested system, that demonstrate non-toxicity for non-cancerous cells, in the case of doxorubicin, cardiac cells would be well appreciated and would increase the evaluation of the efficiency of the system.
2. Furthermore, in vivo studies need further improvements:
Have the authors data about tumour size after treatment (48h)?
Have the authors data about the animal's body weight following the treatment (48h)?
Have the authors dosed systemic markers of cardiac and liver damage after treatment (48h)?
These data are necessary to demonstrate the real efficiency of the system to selectively convey Dox into the cancer site and reduce its side effects.
Minor Points:
1. Specific references are missing in the text after sentences. Please introduce it.
2. Some words are not correct or missing in the text. The authors must rectify these.
3. In materials and methods, the part concerning the tested formulation must be improved in order to clarify the text.
Author Response
Authors’ response to reviewer:
1. The most important complication in the use of Doxorubicin is the toxicity induced in non-cancerous cells. In fact, the authors show in vivo the bioavailability of the micelles predominantly only in the tumor and not in the other organs. In any case, the authors must show in vitro experiments, of tested system, that demonstrate non-toxicity for non-cancerous cells, in the case of doxorubicin, cardiac cells would be well appreciated and would increase the evaluation of the efficiency of the system.
The authors’ answer: Thank you for the valuable comment for the improvements of our manuscript. Indeed, the toxicity and side effect of Doxorubicin induced in non-cancerous cells seriously restricts its clinical application. As shown In vivo biodistribution studies of our origin manuscript, the pHendoSM/DIR micelles showed typical permeability and retention (EPR) mediated tumor-targeting in the mice. To further verify tumor targeting delivery of Dox by the micelles, the distribution comparison of Dox after intravenously injected Dox solution, pHendoSM/Dox and pHendoSM/Dox/Res has been included in the revised manuscript. Compared with the free Dox solution group, the micellar groups showed much stronger tumor targeting and lower cardiac Dox distribution. Therefore, micelles can effectively reduce the side effects of doxorubicin via changing its in vivo distribution.
2. Furthermore, in vivo studies need further improvements:
Have the authors data about tumour size after treatment (48h)?
Have the authors data about the animal's body weight following the treatment (48h)?
Have the authors dosed systemic markers of cardiac and liver damage after treatment (48h)?
These data are necessary to demonstrate the real efficiency of the system to selectively convey Dox into the cancer site and reduce its side effects.
The authors’ answer: Thank you for the valuable comment to the manuscript. We apologized for not providing the in vivo antitumor results (like tumor size, animal's body weight). Currently, our animal lab is under construction for improvement and it is not available for us to conduct the experiment. Furthermore, we only have ten days to revise the manuscript and resubmitted it to the editor office. Even though the in vivo antitumor data are missing from the manuscript, the targeting delivery of Dox and enhanced distribution of Dox in the tumor tissue can be clearly observed in the in vivo distribution study (Fig.8C). The enhanced anti-tumor efficacy and reduced Dox related cardiotoxicity can be anticipated.
3. Specific references are missing in the text after sentences. Please introduce it.
4. Some words are not correct or missing in the text. The authors must rectify these.
The authors’ answer to recommendations both 3 and 4: Thanks to the careful reading of the manuscript. We have carefully examined the missed references, spelling mistakes and typing mistakes in the manuscript. All the corrections made to the manuscript have been highlighted in the revised manuscript.
5. In materials and methods, the part concerning the tested formulation must be improved in order to clarify the text.
The authors’ answer: We apologized for the misunderstanding as we did not clearly specify the experimental tested formulation in “Materials and methods” section. As you recommended, we specified this parts in “Materials and methods” section in revised manuscript. Take one for specific explanation, in the 2.11 Mitochondrial membrane potential detection, the fresh culture medium group was used as the negative group. CCCP group (Mitochondrial electron transfer chain inhibitor) was selected as the positive control group. Dox and Dox/Res were used as the control group and the PHendoSM/Dox/Res was the experimental group. (line 210- line 211)
Round 2
Reviewer 2 Report
Dear Authors,
The paper has improved significantly, and my concerns have beed addressed.
I recommend the paper for acceptance and publication.
Author Response
Dear reviewer:
The sincere appreciation should be given you for your recognition to our revised manuscript.
Best regards
Mingxi Qiao
Reviewer 4 Report
I appreciated the authors' replies but they are not enough to publish the article in this journal. Indeed, concerning cardiotoxicity no experimental response was provided on cardiac cells. Of note, a low distribution of doxorubicin in the heart does not exclude a cardiotoxic effect.
Concerning the anticancer effects the requested data has been provided in minimal part.
Author Response
Authors’ response to reviewer:
1. I appreciated the authors' replies but they are not enough to publish the article in this journal. Indeed, concerning cardiotoxicity no experimental response was provided on cardiac cells. Of note, a low distribution of doxorubicin in the heart does not exclude a cardiotoxic effect. Concerning the anticancer effects the requested data has been provided in minimal part.
The authors’ answer: Thank you for the valuable comment to our manuscript. We apologized for not providing the in vivo antitumor results in R1 version of the manuscript. We didn’t have enough time to finish the experiment at that time. As you recommended, we provided “In Vivo antitumor Efficacy and safety evaluation” in revised manuscript for demonstrating the real efficiency of the system to selectively deliver Dox into the cancer site, enhance the antitumor effects, and reduce its side effects (cardiotoxicity, hepatotoxicity and nephrotoxicity). All the improvements made to the manuscript have been highlighted in the revised manuscript (line 251- line 262 and line 457- line 479).
Round 3
Reviewer 4 Report
Very compliments to the authors, in fact, also without cellular experiments and with a more careful experimental analysis of their in vivo model they provided evidences about the enhanced antitumor effects, and reduced side effects of the system. Now the manuscript is sufficiently revised.